# Pendred Syndrome, or Not Pendred Syndrome? That Is the Question

**DOI:** 10.3390/genes12101569

**Published:** 2021-10-01

**Authors:** Paola Tesolin, Sofia Fiorino, Stefania Lenarduzzi, Elisa Rubinato, Elisabetta Cattaruzzi, Lydie Ammar, Veronica Castro, Eva Orzan, Claudio Granata, Daniele Dell’Orco, Anna Morgan, Giorgia Girotto

**Affiliations:** 1Department of Medicine, Surgery and Health Sciences, University of Trieste, 34149 Trieste, Italy; giorgia.girotto@burlo.trieste.it; 2Department of Life Sciences, University of Trieste, 34127 Trieste, Italy; sofia.fiorino@studenti.units.it; 3Institute for Maternal and Child Health—IRCCS, Burlo Garofolo, 34127 Trieste, Italy; Stefania.lenarduzzi@burlo.trieste.it (S.L.); elisa.rubinato@burlo.trieste.it (E.R.); elisabetta.cattaruzzi@burlo.trieste.it (E.C.); lydie.ammar@burlo.trieste.it (L.A.); veronica.castro@burlo.trieste.it (V.C.); eva.orzan@burlo.trieste.it (E.O.); claudio.granata@burlo.trieste.it (C.G.); anna.morgan@burlo.trieste.it (A.M.); 4Department of Neurosciences, Biomedicine and Movement Sciences, Section of Biological Chemistry, University of Verona, 37129 Verona, Italy; daniele.dellorco@univr.it

**Keywords:** Pendred syndrome, Whole-Exome Sequencing, genotype-phenotype correlation

## Abstract

Pendred syndrome (PDS) is the most common form of syndromic Hearing Loss (HL), characterized by sensorineural HL, inner ear malformations, and goiter, with or without hypothyroidism. *SLC26A4* is the major gene involved, even though ~50% of the patients carry only one pathogenic mutation. This study aims to define the molecular diagnosis for a cohort of 24 suspected-PDS patients characterized by a deep radiological and audiological evaluation. Whole-Exome Sequencing (WES), the analysis of twelve variants upstream of *SLC26A4*, constituting the “CEVA haplotype” and Multiplex Ligation Probe Amplification (MLPA) searching for deletions/duplications in *SLC26A4* gene have been carried out. In five patients (20.8%) homozygous/compound heterozygous *SLC26A4* mutations, or pathogenic mutation *in trans* with the CEVA haplotype have been identified, while five subjects (20.8%) resulted heterozygous for a single variant. *In silico* protein modeling supported the pathogenicity of the detected variants, suggesting an effect on the protein stabilization/function. Interestingly, we identified a genotype-phenotype correlation among those patients carrying *SLC26A4* mutations, whose audiograms presented a characteristic slope at the medium and high frequencies, providing new insights into PDS. Finally, an interesting homozygous variant in *MYO5C* has been identified in one patient negative to *SLC26A4* gene, suggesting the identification of a new HL candidate gene.

## 1. Introduction

Hearing loss (HL) is the most common sensory disorder affecting about 1–3 children in over 1000 [1,2,3]. In the developed countries, the most prevalent causes of HL are genetic defects that impair the production of protein with essential roles in the hearing pathway. HL is generally classified in non-syndromic HL (NSHL), affecting at least 70% of the patients, and syndromic HL (SHL), when other organs abnormalities are present alongside the HL [2]. To date, over 400 syndromes associated with HL have been described [4,5], suggesting the high degree of phenotypic and genetic heterogeneity that characterized the affected patients.

Pendred syndrome (PDS) is considered the most common form of SHL accounting for ~4–10% of all cases of hereditary deafness [6,7]. PDS is inherited with an autosomal recessive pattern and it is associated with sensorineural HL, inner ear malformations, goitre, abnormal organification of iodide, which can ultimately cause hypothyroidism in some subjects, and vestibular dysfunction. 

In detail, PDS patients always display enlarged vestibular aqueduct (EVA) that may or may not be associated with cochlear hypoplasia (when the expected 2.75 turns are reduced to 1.5) with a cystic apex, known as incomplete partition type II (IP-II) [8]. These cochlear abnormalities are also named Mondini malformation. Moreover, EVA is generally detected together with the enlargement of the endolymphatic sac (ESE) and duct [6]. 

As regards to goitre, this feature is not consistently present [9]. In fact, it can be detected in 30–75% of the subjects affected by PDS and becomes apparent after the age of ten years [10,11] with remarkable differences also among individuals of the same kindred [6,9,12]. Indeed, some patients are unaffected, while others present progressive thyroid enlargement leading to multinodular goiter [6]. This observed variability may be attributed to environmental factors, such as nutritional iodine intake, or methods used to assess thyroid enlargement [6,7].

From the audiological point of view, PDS patients usually display prelingual HL, even though in some cases, it can appear later in childhood. In the majority of the patients, the HL is profound, bilateral, progressive and fluctuating, although asymmetry can be present in some cases.

The combination of hearing impairment, EVA or Mondini malformation, and eventually goitre/thyroid dysfunctions is the element that leads to the suspect of PDS as the underlying aetiology of a patient’s HL.

The major gene involved in the pathogenesis of PDS is *SLC26A4,* which consists of 21 exons and is located on chromosome 7. Biallelic mutations in this gene are also responsible for nonsyndromic EVA, DFNB4, characterized by sensorineural HL usually congenital and often severe to profound, vestibular dysfunction, and temporal bone abnormalities (bilateral EVA with or without cochlear hypoplasia).

*SLC26A4* encodes for a member of the solute carrier family 26A named pendrin, an anion exchanger of HCO^3−^, Cl^−^, and I^−^. Pendrin acts as a chloride/bicarbonate exchanger in the inner ear, particularly at the endolymphatic duct and sac level, playing an essential role in the maintenance of inner ear endolymph homeostasis. Furthermore, in the thyroid, it appears to mediate the efflux of iodide from thyroid follicular cells into the follicular lumen.

In less than 1% of the cases, mutations in other genes, such as *FOXI1*, *KCNJ10,* or *EPHA2*, might be responsible of PDS [13,14]. Moreover, a shared haplotype called CEVA (Caucasian EVA) has been recently described as a recessive mutant allele when present *in trans* to a pathogenic variant of *SLC26A4*. The haplotype comprises 12 variants located in introns or intergenic regions upstream of *SLC26A4* and is generally identified in Caucasian patients [15]. 

Despite technological advances, current genetic tests allow one to provide a molecular diagnosis for only 25–50% of the PDS patients [12,13,15]. Generally, the patients are classified as M0 when no variant is highlighted, M1 when only one mutation is detected, and M2 when both mutant alleles are identified [15]. This consideration suggests that other genetic contributions to the manifestation of the disease remain hidden from the routinely performed assays. 

Here we describe a selected cohort of 24 deeply characterized patients with a hypothesized clinical diagnosis of PDS analyzed by Whole Exome Sequencing (WES) and Multiplex Ligation Probe Amplification (MLPA), together with the evaluation of the CEVA haplotype with the aims to: (1) identify the genetic variants responsible of the patients’ disorder and (2) define a genotype-phenotype correlation taking advantage of the highly detailed phenotypic data (both audiological and radiological).

## 2. Materials and Methods

### 2.1. Samples Collection

Enrolled patients, all of Caucasian origin recruited at the Institute for Maternal and Child Health - I.R.C.C.S. “Burlo Garofolo” (Trieste, Italy) provided written informed consent and their next of kin provided it in case of minors. All research was conducted according to the ethical standard defined by the Helsinki Declaration.

The individuals underwent a careful clinical evaluation. In detail, patients’ past medical records were investigated by clinical geneticists to exclude HL cases that could be ascribed to non-genetic causes (e.g., infections or trauma). The patients underwent a careful clinical evaluation specifically focused on dysmorphic features and other congenital abnormalities not specifically related with the syndrome itself. Moreover, for all the patients, pure tone audiometric testing (PTA) or auditory brainstem response (ABR) was performed to define the degree of HL. To define the presence of EVA the Cincinnati criteria were followed (e.g., the vestibular aqueduct is defined enlarged if it has a midpoint width (1.0 mm) or opercular width (2.0 mm) greater than the 95th percentile) [16,17]. Thus, to perform the statistical analysis Magnetic Resonance Imaging (MRI) and Computerized Tomography (CT) scan were performed as well. Finally, thyroid function assessments were carried out. The levels of serum thyroid stimulating hormone (TSH), serum free thyroxine (T4), and serum free triiodothyronine (T3) were evaluated. Ultrasound was also performed whenever altered levels were found on the blood test or if thyroid clinical examination suggested the presence of goiter or nodules. The thyroid was considered enlarged if the measurements reveled a mean volume greater than 8 to 10 mL (range 3 to 20 mL) [18,19].

### 2.2. DNA Extraction and Quantification

Genomic DNA was extracted from peripheral whole blood samples using the QIAsymphony® SP instrument with QIAsymphony® DNA Midi kit (Qiagen, Venlo, The Netherlands) and DNA concentration measured using Nanodrop ND 1000 spectrophotometer (NanoDrop Technologies Inc., Wilmington, DE, USA).

### 2.3. GJB2 and GJB6 Analyses 

Sanger method was employed to sequence the entire coding region of *GJB2* gene (primers available upon request). In particular, DNA was analyzed on a 3500 Dx Genetic Analyzer (ThermoFisher, Waltham, MA, USA), using ABI PRISM 3.1 Big Dye terminator chemistry (ThermoFisher, Waltham, MA, USA) according to the manufacturer’s instructions. Subsequently, *GJB6* deletions (i.e., D13S1830- D13S1854) were screened by multiplex PCR, as previously described [20].

### 2.4. Whole Exome Sequencing (WES) 

WES was carried out on an Illumina NextSeq 550 instrument (Illumina Inc., San Diego, CA, USA). According to the manufacturer’s instructions. Briefly, 50 ng of genomic DNA were enzymatically fragmented, and, after an end repair and dA-tailing reaction, each fragment was ligated to a universal adapter and then amplified using the Unique Dual Index primers (Twist Bioscience). Afterwards, genomic libraries were prepared using the Twist Human Core Exome + Human RefSeq Panel kit (Twist Bioscience, South San Francisco, CA, USA) that allows to cover 99% of the protein-coding genes. In conclusion, the hybridized fragments have been captures, amplified and sequenced.

FASTQ files were processed through a custom pipeline (*Germline-Pipeline)*, developed by enGenome srl (https://www.engenome.com/ (accessed on 7 September 2021), which includes FASTQ trimming, FASTQ Quality Check, FASTQ Mapping, Mark of Duplicates, Base Quality Score Recalibration, and Variant Calling.

This workflow, designed for Illumina paired-end sequencing data, allows the generation of a final VCF file containing information regarding germline variants, such as Single Nucleotide Variants (SNVs), short insertion/deletions (INDELs) and exon-level copy number variations (CNVs). Finally, VCF files were analyzed on EnGenome Expert Variant Interpreter (eVai) software (evai.engenome.com) that allows variant annotation and interpretation. In particular, eVai combines Artificial Intelligence with the American College of Medical Genetics (ACMG) guidelines [21] to classify and prioritize every genomic variant, suggesting a list of possible related genetic diagnoses.

SNVs and INDELs were excluded if they lead to synonymous amino acid substitutions that were not predicted as damaging or did not affect splicing or highly conserved residues. Furthermore, variants with a quality score (QUAL) < 20 or called in off-target regions were excluded as well. 

Variants previously reported as polymorphism were removed thanks to a comparison between the identified genetic variants and data reported in NCBI dbSNP build153.

(http://www.ncbi.nlm.nih.gov/SNP/ (accessed on 7 September 2021) as well as in gnomAD (http://gnomad.broadinstitute.org/ (accessed on 7 September 2021). In particular, a Minor Allele Frequency (MAF) cutoff of 0.001 was used. 

The pathogenicity of known genetic variants was evaluated using ClinVar (http://www.ncbi.nlm.nih.gov/clinvar/ (accessed on 7 September 2021), Deafness Variation Database (http://deafnessvariationdatabase.org/ (accessed on 7 September 2021), and The Human Gene Mutation Database (http://www.hgmd.cf.ac.uk/ac/ index.php (accessed on 7 September 2021). 

Several *in silico* tools, such as PolyPhen-2 [22], SIFT [23], Pseudo Amino Acid Protein Intolerance Variant Predictor (for coding variants SNVs/INDELs) (PaPI score) [24], Deep Neural Network Variant Predictor (for coding/non-coding variants, SNVs) (DANN score) [25] and dbscSNV score [26] were used to evaluate the pathogenicity of novel variants. 

Finally, on a patient by patient basis, variants were discussed in the context of phenotypic data at interdisciplinary meetings. As the last step, and the most likely disease-causing variants were confirmed by direct Sanger sequencing.

### 2.5. CEVA Haplotype Analysis

Considering that the subjects involved in the study are Caucasian, we employed Sanger sequencing to test the presence of the CEVA haplotype in M1 patients. We followed the procedure as already described in 2.3. (primers available upon request). In particular, we sequenced fragment of 200–400 bp containing the twelve SNP of the haplotype (i.e., rs17424561, rs79579403, rs17425867, rs117113959, rs17349280, rs117386523, rs80149210, rs199667576, rs9649298, rs117714350, rs199915614, rs150942317).

### 2.6. Multiplex Ligation Probe Amplification (MLPA) 

MLPA analysis was performed searching for deletion/duplication in the *SLC26A4* gene. The SALSA^®^ MLPA^®^ probe mixes P280-100R SLC26A4 (MRC-Holland, Amsterdam, the Netherlands) was employed, according to the manufacturer’s protocol. For the data analysis, the software Coffalyser.Net was used in combination with the lot-specific MLPA Coffalyser sheet. The following cutoff values for the dosage quotient (DQ) of the probes were used to interpret MLPA results: 0.80 < DQ < 1.20 (no deletion/duplication), DQ = 0 (deletion), and 1.75 < DQ < 2.15 (duplication). 

### 2.7. Prediction of Membrane Topology and 3D Molecular Model of Pendrin

TOPCONS [27] was used to predict the consensus prediction of membrane protein topology of pendrin (encoded by *SLC26A4*). No homologous transmembrane protein was detected in the Protein Data Bank. Overall, the software predicted 13 transmembrane (TM) regions, 7 intracellular regions and 6 extracellular regions (Figure 1) with different confidence.

A BLASTP run of the sequence of human pendrin using the Protein Data Bank as a reference database highlighted some hits, the best one covering 91% of the protein sequence (human SLC26A9 chain A; PDB: 7CH1; s.i. 35.13%). We also used the neural network based Alphafold algorithm [28] for the prediction of human Pendrin three-dimensional structure (https://alphafold.ebi.ac.uk/entry/O43511 (accessed on 7 September 2021). The performance was satisfactory, especially for the TM regions, for which very high (predicted Local Distance Difference Test, pLDDT < 90) to confident (70 < pLDDT < 90) levels characterized the prediction of the TM regions. Superposition of the experimental (CryoEM) structure of chain A of the dimeric SLC26A9 assembly (PDB: 7CH1) [29] to the structure originated by Alphafold led to relatively low (2.73 Å) root-mean squared deviation (RMSD); therefore, we used the Alphafold model for the analysis of the putative effects of amino acid substitutions in pendrin.

## 3. Results

Twenty-four patients presenting with HL and temporal bone abnormalities suggesting PDS have been considered in the study. 

All the clinical features of the enrolled patients are reported in Table 1. CT and MRI were performed in 17 out of 24 patients, while CT alone was carried out for 23 patients out of 24. Briefly, the in-depth morphological evaluation highlighted EVA, ESE, Mondini malformation, or IP-II, and semicircular canal and vestibular abnormalities in the vast majority of them. In particular, EVA, and eventually ESE, alone were present in 21% of our patients (ID6, ID8, ID12, ID18, ID23). On the other hand, both EVA and Mondini malformation were detected in 79% of the patients (ID1, ID2, ID3, ID4, ID5, ID7, ID9, ID10, ID11, ID13, ID14, ID15, ID16, ID17, ID19, ID20, ID21, ID22, ID24). As regards thyroid function tests (serum TSH, serum-free T4, and serum-free T3 levels), data were available for 23 out of 24 patients and one (ID6) showed subclinical hypothyroidism (TSH 5.07 mU/L with a normal value ranging from 0.27 and 4.20) at the age of 5 years. Ultrasound examination of the thyroid was performed in two out of 24 patients.

All the patients underwent a first round of genetic testing, including the analysis of *GJB2* and *GJB6* genes, which resulted negative. 

Subsequently, WES was carried out, first focusing on genes already associated with PDS, such as *SLC26A4*, *FOXI1*, *KCNJ10* [13], and *EPHA2* [14]. Afterwards, the CEVA haplotype in M1 individuals has been checked. 

WES allowed to identify the molecular cause of PDS in four patients. 

In particular, WES highlighted the presence of homozygous and compound heterozygosis mutations in the *SLC26A4* (NM_000441.1) gene in four patients:Patient ID1: Homozygous for c.1225C>T, p.(R409C)Patient ID2: Homozygous for c.704A>G, p.(Q235R)Patient ID22: Homozygous for c.1489G>A, p.(G497S)Patient ID12: Compound heterozygous for c.1001+1G>A and c.1149+3A>G

In detail, ID1, a 29-year-old female, is affected by profound HL, with a notable worst performance at medium and high frequencies (Figure 2), together with EVA, Mondini, ESE, semicircular canal/vestibular abnormalities, and no thyroid function dysfunction. 

WES revealed the presence of a homozygous mutation (c.1225C>T, p.(R409C)) predicted as pathogenic by several *in silico* tools and already described as pathogenic in association with EVA [30] (Table 2). 

Patient ID2, a 2-year-old male, displays profound congenital HL (Figure 2) together with bilateral EVA, ESE, IP-II and semicircular canal/vestibular abnormalities. Regarding his thyroid status, TSH, T3, and T4 levels were normal at the age of 13 months. The patient carries a homozygous mutation (c.704A>G, NM_000441.1, p.(Q235R) (Table 2) that has already been classified as pathogenic and associated with bilateral EVA [32].

Patient ID22, a 29-year-old female, shows an audiometric pattern characterized by a profound slope at the medium and high frequencies, and mild/moderate HL at the low ones (Figure 2) in addition to bilateral EVA, ESE, IP-II and semicircular canal/vestibular abnormalities (Table 1). At the age of 28, her last thyroid ultrasound showed mild enlargement of the thyroid gland with a heterogeneous echogenicity without nodules. TSH, T3 and T4 levels have been normal throughout her follow-up. ID22 carries a homozygous mutation (c.1489G>A, NM_000441.1, p.(G497S)) classified as pathogenic and detected in patients displaying HL and EVA [34] (Table 2).

A molecular diagnosis was also provided for patient ID12, a 12-year-old female who carries two *SLC26A4* mutations at the compound heterozygous state. The patient displays moderate and asymmetric HL at the low frequencies and severe/profound HL at the medium and high frequencies (Figure 2) together with bilateral EVA (Table 1) without thyroid dysfunction. The two known mutations (c.1001+1G>A and c.1149+3A>G), already classified as pathogenic, alter the splicing process [35,36] (Table 2). 

Moreover, six additional patients resulted carriers of only one heterozygous mutation and were initially classified as M1:ID7: Heterozygous c.1536_1537delAG, p.(R512Sfs*14)ID10: Heterozygous c.1489G>A, p.(G497S)ID14: Heterozygous c.554G>C, p.(R185T)ID17: Heterozygous c.1263+2T>CID18: Heterozygous c.1730T>C, p. (V577A)ID19: Heterozygous c.600G>A, p.(Q200Q)

All the variants were classified as pathogenic by different *in silico* prediction tools and have been previously associated with PDS or EVA [30,33,34,35,36,37,38], with the only exception of the one carried by ID19 (Table 2). All the patients, aged between six and 16, are affected by moderate to profound HL and displayed both bilateral EVA and IP-II. The only exception is ID10, who displays asymmetrical HL and monolateral EVA and IP-II, all affecting mainly the left ear (Table 1, Figure 2, Appendix A). For all six patients, we were able to measure TSH, T3, and T4 levels and so far, no one showed hypothyroidism or subclinical hypothyroidism. However, one patient, ID17, had a thyroid ultrasound showing a normal thyroid volume associated with a heterogeneous echogenicity of the thyroid gland without focal lesions. 

Interestingly, for patient ID19, a 16-year-old male, who displays profound bilateral HL, EVA, IP-II and semicircular canal/vestibular abnormalities, the analysis of the CEVA haplotype revealed its presence *in trans* with the pathogenic variant ((c.600G>A, NM_000441.1, p.(Q200Q))) (Table 2). The predicted effects of this allele on the encoded protein are both synonymous aminoacidic substitution and splicing alteration; in fact, the nucleotide change involved the last nucleotide of exon 5, likely altering the normal splicing site and thus impacting on the final structure of the pendrin protein. 

Further, we performed MLPA analysis targeting the *SLC26A4* gene in the M1 and M0 patients, and all the individuals resulted negative for deletion or duplication in the *SLC26A4* gene.

In order to strengthen the pathogenic effect of the identified variants, we performed an *in silico* protein modeling. To date, no experimental structure has been solved for pendrin. The performance of the novel neuronal network-based algorithm Alphafold [28] is extremely high in predicting the three-dimensional fold of proteins even when the sequence similarity is too low to allow for the building of reliable homology models. The Alphafold-predicted three-dimensional structure of pendrin is reported in Figure 3.

The overall topology is in line with predictions based on the widely used TOPCONS methodology (see methods) and shows the typical fold of a transmembrane transporter. It is worth noting that the protein sequence of pendrin shares 32% sequence identity with the human SLC26A9 Cl^2212^ and Na^+^ transporter, for which the dimeric cryo-EM structure has been recently solved [29]. The structure represented in Figure 3 is therefore a prediction limited to the monomeric subunit of a putatively dimeric assembly. 

All mutated sidechains in the missense variants found in this study except V577A, located in the cytoplasmic domain, are in the transmembrane region of pendrin and their predicted orientation is towards the interior of the transporter rather than towards the lipid milieu (Figure 3A, B, viewed from the top). It is therefore plausible that the mutations do not affect the supramolecular assembly of pendrin into dimers, but rather may lead to destabilization of the protein or prevent its function of sodium-independent transporter of chloride and iodide.

Interestingly, some pendrin mutations identified here correspond to residues playing important functional roles in SLC26A9. In particular, Arginine in position 185, substituted by a Threonine in p.(R185T) is the analog of R169 in SLC26A9, a basic residue involved in an electrostatic interaction with a Cl^2212^ ion. Moreover, the side chain of R409 protrudes to a region that, in SLC26A9 is occupied by a Na^+^ ion and the R409C substitution would lose any potentially conserved electrostatic intersection in pendrin. It is not clear what could be the effect of the V577A substitution in the cytoplasmic milieu of pendrin, but the fold of that region is predicted with less confidence by Alphafold and structural differences with SLC26A9 are more prominent that in other regions. As for the other variants, the G497S substitution might induce some steric clash between adjacent helices, while the Q235R substitution might induce electrostatic repulsion with K447, thus destabilizing the protein core. Finally, replacing the aliphatic L445 side chain with a bulky, aromatic Trp could also lead to significant structural rearrangement in a packed hydrophobic region. As for the p.(Q200Q) variant, the synonymous mutation is obviously not predicted to affect either protein structure or function.

Finally, 14 individuals did not display any causative mutation in *SLC26A4* (M0), nor in other PDS related genes. The comparison between their audiometric profiles with those of M2/M1 individuals revealed a milder phenotype in M0 patients, suggesting the possible involvement of other causative genes. (Appendix A). In this light, we searched for variants in genes already known for being causative of SHL/NSHL (Hereditary Hearing Loss Homepage; http://hereditaryhearingloss.org/, date of access: 7 September 2021), or in genes likely involved in the auditory function, without finding any interesting result, with the only exception of patient ID24. The subject carries an interesting homozygous variant in *MYO5C* (NM_018728.4, c.3592C>T p.(R1198C)) (Table 2), a novel gene that has not been associated with HL yet. The variant is predicted as pathogenic by the *in silico* tools used during data analysis and is described with an ultra-rare minor allele frequency in the gnomAD database (rs200717531), with no homozygous individuals reported. ID24 displays bilateral EVA, without ESE, bilateral IP-II, semicircular canal and vestibular abnormalities (Table 1). The newborn hearing screening revealed a monolateral hearing impairment (right ear), that progressed rapidly to bilateral and profound HL (Figure 4) by the age of three. The patient did not manifest hypothyroidism or subclinical hypothyroidism.

## 4. Discussion

When PDS was first described, the diagnosis was made only based on clinical signs (e.g. deafness, goitre, thyroid dysfunctions, temporal bone malformations). However, since the discovery of the main disease-causing gene, *SLC26A4,* the diagnostic procedures have improved, and today it is possible to define the molecular basis of the patients’ disease [9]. Nevertheless, PDS patients are characterized by high clinical variability and the detection rates of PDS causing mutations remain limited, highlighting the importance of a careful clinical examination.

According to literature data, the detection rate of at least one mutation ranges between 25–50% [12,13,15]. Among our cohort, we identified five M2 patients (20.8%) (including one patient carrier of a pathogenic mutation *in trans* with the CEVA haplotype) and five M1 subjects (20.8%), for an overall *SLC26A4*-mutations carriers rate of 41.7%. All the mutations here identified are predicted as damaging and our *in silico* modeling suggested their possible effect on protein stabilization and function.

The lack of a second pathogenic allele explaining the clinical phenotype of M1 patients might be due to several reasons, some of them tested in our work.

In a limited series of patients, the hypothesis of a digenic inheritance involving other genes, such as *FOXI1*, *KCNJ10,* and *EPHA2*, has been shown [13,14]; however, none of the subjects of our cohort carried a likely pathogenic variant in any of the above-mentioned genes. 

We also checked for the presence of any CNV in the *SLC26A4* gene using MLPA, with negative results. In this light, after excluding the presence of the CEVA haplotype, the most reasonable explanation implies the presence of variants in regions not explored with WES, or complex structural variants not detectable with the methodologies used in this work. In addition, a new pathophysiological model for PDS has also been recently hypothesized [39]. Hosoya et al. proposed that a single mutation on the *SLC26A4* gene might cause the formation of misfolded pendrin aggregates and, if defects of the ubiquitin-proteasome system and/or autophagy pathways are also present, this could ultimately result in inner ear cells death. Briefly, they suggest that some M1 patients’ phenotype might be due to a decreased clearance of the mutant protein causing degeneration, rather than a channelopathy. Thus, specific genetic backgrounds, but also environmental factors, might exacerbate the stress induced by monoallelic *SLC26A4* mutation leading to inner ear damage, which could ultimately cause the PDS phenotype in a specific subset of patients. 

Based on these data, additional efforts for understanding the pathological mechanism responsible for the clinical phenotype of these patients are needed, including, for instance, the application of whole genome sequencing, RNA studies to detect quantitative or qualitative defects in *SLC26A4* transcripts or further studies involving human-induced pluripotent stems cells from PDS patients. 

Given the availability of an extreme accurate clinical evaluation, we tried to outline the main features of our PDS patients.

As regards the radiological observations, a clear distinction between M2/M1 and M0 patients could not be made. Thus, the identification of bilateral EVA or Mondini should not be considered as a proof of the presence of *SLC26A4* gene variants.

Data regarding the thyroid hormone levels were available for almost all the patients (23 out of 24) and except for one patient (ID6, 13-year-old) who shows subclinical hypothyroidism, all the other subjects seem to be euthyroid. As mentioned before, goitre generally becomes apparent in adolescence (more or less after the age of ten) and in our cohort ten out of 23 patients are aged <10 years. Moreover, environmental factors, such as iodine intake can play an important role since literature data report that the prevalence of goitre is inversely proportional to daily iodine intake [40]. Considering that all the recruited patients live in Trieste, a seaside city, it is reasonable to hypothesize that they are exposed daily to high iodine concentrations.

Concerning the audiological data, interestingly, we highlighted a new genotype-phenotype correlation that, to our knowledge, has been previously described but never deeply explored due to a lack of accurate audiological data of PDS patients [3,40,41,42,43,44]. In particular, M1 and M2 subjects display highly similar audiometric patterns. The majority of these patients are characterized by mild-moderate HL at the low frequency, which gradually worsens, becoming moderate-severe at the medium frequencies and profound at the high frequencies. Only few M1/M2 patients display profound HL with flat audiograms. Furthermore, audiometric data of M1/M2 patients seem to differ from the M0 cases who show a milder phenotype, confirming the hypothesis that different genetic causes should be searched in these patients. Furthermore, we identified an unexpected phenotype in a M2 patient carrier of the CEVA haplotype. In fact, according to literature data the concomitant presence of a heterozygous mutation *in trans* with CEVA haplotype is associated with milder forms of PDS [15]. However, patient ID19 displays a profound hearing impairment at all frequencies, suggesting a higher phenotypic variability associated with the CEVA haplotype. 

The remaining 14 patients of our cohort do not carry any mutation in PDS genes, nor in other known HL genes. Moreover, eight out of 14 M0 patients displayed fluctuation and/or progression of the HL, which are signs typically associated with PDS. This consideration might suggest that the HL of these eight patients could be ascribed to mutations located in genomic regions not explored with WES, such as *SLC26A4* introns. Alternatively, the causative mutations might be found in novel disease genes. Interestingly, among the M0 individuals, we identified a patient carrier of a homozygous missense variant in *MYO5C* gene. *MYO5C* encodes the myosin-Vc protein that belongs to the class V myosins, one of the most ancient and distributed group of myosins, hypothesized to act as motors for actin-dependent organelle transport [45]. Myosin-Vc seems to be mostly expressed in epithelial and glandular tissues and shares ~50% identity with the two other classes of myosins V in vertebrates, myosin-Va (Myo5a) and myosin-Vb (Myo5b) [46]. Even though no mutations in *MYO5C* have been linked to any known disease yet, other myosins have been associated with HL before, such as *MYO15A*, *MYO3A*, *MYO6*, *MYO7A*, etc [47]. In this light, the identification of an ultra-rare homozygous missense variant, predicted as pathogenic by several *in silico* tools, represents an interesting finding worth mentioning. Additional studies clarifying the role of this gene will be crucial to define if it indeed could be involved the etiopathogenesis of HL. Nevertheless, the evaluation of *MYO5C* in other PDS-like subjects may lead to the identification of other patients carriers of pathogenic variant, reinforcing the hypothesis of a novel candidate gene for PDS.

Overall, these findings highlight the efficacy of WES and the analysis of the CEVA haplotype together with an accurate clinical evaluation, both from the audiological, and radiological point of view. In fact, a deep phenotypical evaluation might allow a precise definition of the essential symptoms common to all PDS patients leading to the definition of consistent phenotype-genotype correlations that will ultimately increase the detection rates. 

## Figures and Tables

**Figure 1 genes-12-01569-f001:**
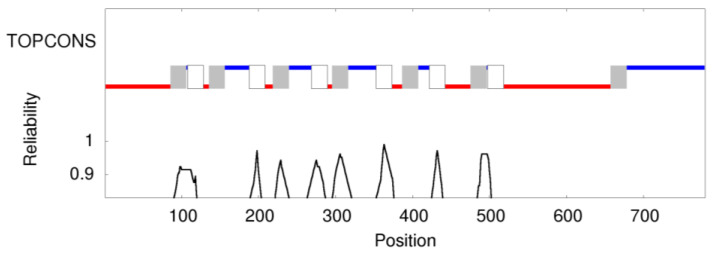
Membrane topology prediction for the protein expressed by *SLC26A4* (pendrin) according to TOPCONS [27]. Boxes refer to the transmembrane (TM) region (in vs. out, grey; out vs. in, white); red lines refer to intracellular regions; blue lines refer to extracellular regions. X-axis refers to the position within protein sequence. Reliability estimate of the predictions is reported in the Y-axis.

**Figure 2 genes-12-01569-f002:**
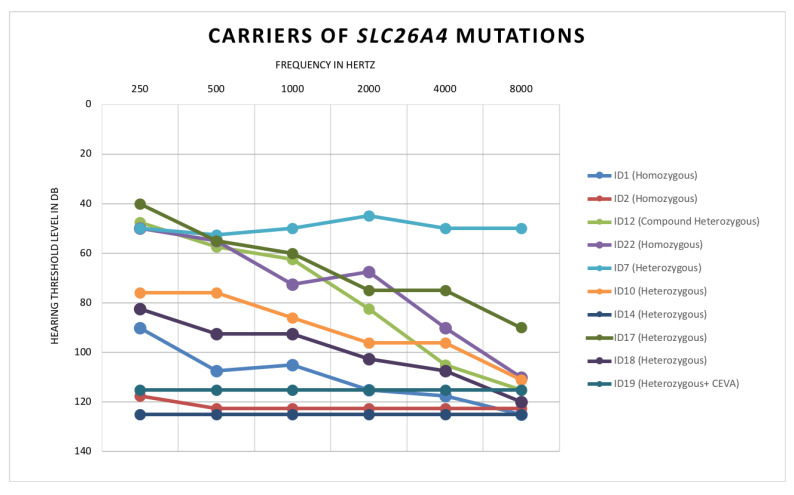
Audiograms of the carriers of *SLC26A4* mutations. Audiometric features, displayed as audiograms, of the ten patients resulted in being carriers of *SLC26A4* homozygous or heterozygous mutations. The hearing threshold levels are reported as the mean value between the right and the left ear.

**Figure 3 genes-12-01569-f003:**
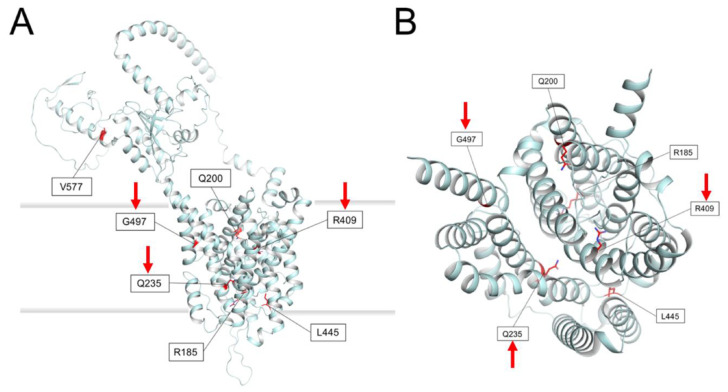
Structural model of the human pendrin monomer as predicted by Alphafold [28]. Protein backbone is represented by cyan cartoons and residues target of missense mutations identified in this study are represented by red sticks and labeled. The red arrows indicate the variants that were detected in homozygosis in the cohort. (**A**) View in a direction crossing the cell membrane. The membrane bilayer is represented by shaded thick gray lines. The cytoplasmic side of the protein is in the upper part while the extracellular portion is in the lower part. (**B**) View of the transmembrane region as seen from the top of the view in (**A**). Protein domains not belonging to the transmembrane region have been omitted for clarity.

**Figure 4 genes-12-01569-f004:**
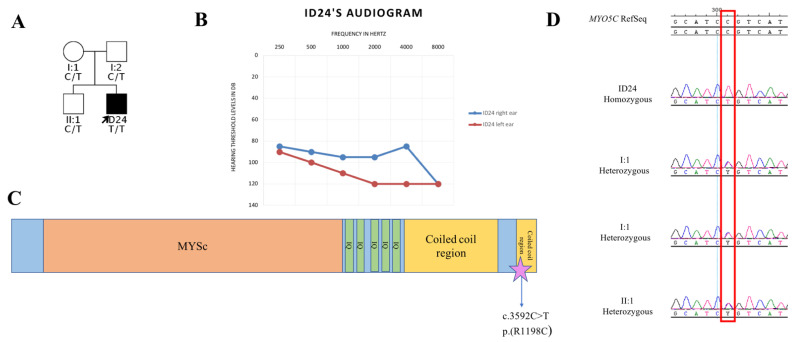
Pedigree and audiograms of patient ID24, DNA chromatograms, and schematic representation of the protein domains (**A**) Pedigree of the proband ID24, which is indicated with a black arrow. Both his parents (I:1 and I:2) and his brother (II:1) carry the variant on *MYO5C* (NM_018728.4, c.3592C>T p.(R1198C)) at the heterozygous state. The genotypes of the four individuals are indicated. The only affected subject is ID24, who therefore is represented with a filled symbol. square: males; circle: females. (**B**) Audiometric features of ID24, displayed as audiograms. The hearing threshold levels of the right ear are reported in blue while those of the left ear are reported in red. (**C**) Schematic representation of myosin-Vc protein domains and the localization of the variant identified in ID24. The MYSc (myosin large ATPase) domain is displayed in orange, in green the five IQ domains, which are formed by short calmodulin-binding motif and finally the coiled coil domains are reported in yellow. A pink star points out the position of the interesting *MYO5C* variant identified in ID24. (**D**) Chromatograms displaying part of *MYO5C* sequence. The sequence of ID24 is reported on top, followed by the one of his mother, father, and brother, respectively. The red box indicates the position of the variant.

**Table 1 genes-12-01569-t001:** Clinical and genetic features of the suspected-PDS patients.

Patient	Age (Years Old)	EVA	Severity of the Malformation	ESE	Cochlear Abnormalities (IP-II)	Semicircular Canal Vestibular Abnormalities	Average Hearing Threshold	Classification
ID1	29	Bilateral	Moderate	Bilateral	Bilateral	Bilateral	Profound	M2
ID2	2	Bilateral	Moderate	Bilateral	Bilateral	Bilateral	Profound	M2
ID3	11	Bilateral	Moderate	Bilateral	Bilateral	Bilateral	Severe	NA
ID4	3	Bilateral	Mild	Bilateral	Bilateral	Bilateral	Mild	NA
ID5	14	Bilateral	Moderate	Bilateral	Bilateral	Bilateral	Moderate	NA
ID6	13	Bilateral	Moderate	Bilateral	NO	Bilateral	Moderate	NA
ID7	12	Bilateral	Moderate	Bilateral	Bilateral	Bilateral	Moderate	M1
ID8	13	Bilateral	Mild	NO	NO	Bilateral	Moderate	NA
ID9	21	Left	Mild	NO	IP-II Left	Bilateral	Moderate	NA
ID10	10	Left	Mild	Left	IP-II Left	NA	Profound	M1
ID11	30	Bilateral	Moderate	Bilateral	Bilateral	Bilateral	Severe	NA
ID12	12	Bilateral	Mild	NA	NA	NA	Severe	M2
ID13	7	Bilateral	Mild	Bilateral	Bilateral	Bilateral	Severe	NA
ID14	6	Bilateral	Mild	NO	Bilateral	Bilateral	Profound	M1
ID15	13	Bilateral	Moderate	Bilateral	Bilateral	NO	Moderate	NA
ID16	6	Bilateral	Moderate	Bilateral	Bilateral	NO	Moderate	NA
ID17	15	Bilateral	Moderate	Bilateral	Bilateral	Bilateral	Moderate	M1
ID18	7	Bilateral	NA	NA	NA	NA	Profound	M1
ID19	16	Bilateral	Moderate	NA	Bilateral	Bilateral	Profound	M2
ID20	10	Bilateral	Mild	NA	Bilateral	Bilateral	Moderate	NA
ID21	23	Bilateral	Mild	Bilateral	Bilateral	Bilateral	Moderate	NA
ID22	29	Bilateral	NA	Bilateral	Bilateral	Bilateral	Severe	M2
ID23	1	Bilateral	Mild	Left	NA	NA	Severe	NA
ID24	13	Bilateral	Moderate	NA	Bilateral	Bilateral	Profound	NA

The table describes the phenotypical characteristics and the genetic findings of all the 24 patients recruited in the present study. EVA: enlarged vestibular aqueduct; ESE: enlarged endolymphatic sac; NA: not available; NO: not detected.

**Table 2 genes-12-01569-t002:** *SLC26A4* gene variants identified in the selected cohort of suspected PDS patients.

Patient	*SLC26A4* cDNA Change	Classification	Protein Change	dbSNP	GnomAD	PolyPhen [22]	SIFT [23]	Mutation Taster [31]	Reference
ID1	c.1225C>T (Hom)	M2	p.(R409C)	rs147952620	0.001594%	1 (d)	0 (d)	d. c.	[30]
ID2	c.704A>T (Hom)	M2	p.(Q235R)	rs752485540	0.0007357%	1 (d)	0 (d)	d. c.	[32]
ID7	c.1536_1537delAG (Het)	M1	p.(R512Sfs*14)	rs1435734312	NA	NA	NA	d. c.	[33]
ID10	c.1489G>A (Het)	M1	p.(G497S)	rs111033308	0.002475%	1 (d)	0 (d)	d. c.	[34]
ID12	c.1001+1G>A (Het)	M2	NA	rs80338849	0.02059%	NA	NA	d.c.	[35]
	c.1149+3A>G (Het)		NA	rs111033314	0.002389%	NA	NA	d.c.	[36]
ID14	c.554G>C (Het)	M1	p.(R185T)	rs542620119	0.008485%	0.98 (d)	0.04 (d)	d. c.	[30]
ID17	c.1263+2T>C (Het)	M1	NA	NA	NA	NA	NA	d.c.	[37]
ID18	c.1730T>C (Het)	M1	p. (V577A)	rs56017519	0.0003982%	0.999 (d)	0.1 (t)	d. c.	[38]
ID19	c.600G>A (Het)	M2 (CEVA)	p.(Q200Q)	NA	NA	NA	NA	d. c.	NA
ID22	c.1489G>A (Hom)	M2	p.(G497S)	rs111033308	0.002475%	1 (d)	0 (d)	d. c.	[34]

Variants identified in ten patients. For each one, the results of three in-silico prediction tools are reported (t = tolerated, d = damaging, d.c. = disease-causing; NA: not available). For the already described mutation, references describing the pathogenicity of the variant are reported.

## Data Availability

The data presented in this study are available on request from the corresponding author. The data are not publicly available due to privacy restrictions.

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
