# Peer review of "Pendred Syndrome, or Not Pendred Syndrome? That Is the Question"

_genes, 2021, doi:10.3390/genes12101569_

Round 1

Reviewer 1 Report

Tesolin and colleagues preformed serial molecular diagnosis for a cohort of 24 subjects with HL and temporal bone abnormalities suggesting PDS. Fiver M2 subjects (20.8%), including one with CEVA haplotype, and 5 M1 subjects (20.8%) were identified, respectively. Also, subjects with M0 presented relatively milder hearing loss level than M2/M1 subjects. Two subjects received thyroid sonography, and one was reported to have mild enlargement of the thyroid gland with a heterogeneous echogenicity without nodules. Regarding to the thyroid function tests, 1 subject showed subclinical hypothyroidism.

In general, this is a well-written article and the authors performed serial molecular diagnosis in their cohort. They also performed comprehensive image studies, audiologic exams and thyroid function tests in EVA cohort. The authors pointed out the genotype-phenotype correlation in the cohort.

  1. The inclusion criteria were not clearly identified. What were the criteria of EVA/ESE? Were the image findings identified from CT or MRI?
  2. Some subjects presented unilateral inner ear malformations. Did they present bilateral hearing loss?
  3. Goiter is more common than hypothyroidism in EVA patients and is the key phenotype in distinguishing PDS from DFNB4. Fifteen subjects in this cohort age more than 10 years, but only 2 received thyroid sonography. What are the criteria of goiter/thyroid enlargement adopted in this study? The limited data of thyroid sonography made difficulties in diagnosis PDS or non-syndromic EVA.
  4. What are the audiologic features of M0 subjects besides milder hearing loss threshold? Did they present progressive and fluctuating hearing levels?
  5. In result part, the nucleotide change of c.704A>T in SLC26A4 does not lead to p.Q235R in aminoacidic change. (Page 8, line 24)
  6. Page 8, table 2. Did the genetic diagnosis of ID22 subject was homozygous c.1489G>A? Did the subject belong to M2 instead of M1 group?
  7. The pathogenicity of MYO5C should be further clarified. And more evidence of the association between MYO5C and HL is needed.

Author Response

Reviewer #1 (Comments to the Author):

Tesolin and colleagues preformed serial molecular diagnosis for a cohort of 24 subjects with HL and temporal bone abnormalities suggesting PDS. Fiver M2 subjects (20.8%), including one with CEVA haplotype, and 5 M1 subjects (20.8%) were identified, respectively. Also, subjects with M0 presented relatively milder hearing loss level than M2/M1 subjects. Two subjects received thyroid sonography, and one was reported to have mild enlargement of the thyroid gland with a heterogeneous echogenicity without nodules. Regarding to the thyroid function tests, 1 subject showed subclinical hypothyroidism.

In general, this is a well-written article and the authors performed serial molecular diagnosis in their cohort. They also performed comprehensive image studies, audiologic exams and thyroid function tests in EVA cohort. The authors pointed out the genotype-phenotype correlation in the cohort.

1.The inclusion criteria were not clearly identified. What were the criteria of EVA/ESE? Were the image findings identified from CT or MRI?

We thank the Reviewer for the comment. The Cincinnati criteria were adopted to define if the patients were affected by EVA/ESE (Sarioglu et al. 2021; Dewan et al. 2009). According to this classification, a vestibular aqueduct is defined as enlarged if it has a midpoint width (1.0 mm) or opercular width (2.0 mm) greater than the 95th percentile. To perform this statistical analysis, CT is essential. Nevertheless, most of the patients underwent both CT and MRI, except patients ID19, ID20, ID21, ID22, ID23 and ID24, for whom only CT was performed. Patient ID18 has been first followed in another institute, thus his imaging results were not available. Nevertheless, we decided to include him in the study because his medical records reported bilateral EVA. We amended the manuscript by adding few lines regarding these aspects (lines 108-112 and 217-219).

2.Some subjects presented unilateral inner ear malformations. Did they present bilateral hearing loss?

We acknowledge the Reviewer for the comment. Patients ID9 and ID10 present unilateral inner ear malformations and asymmetrical hearing loss. In both cases, the malformations affect the left ear, which is also the ear with the worst hearing performance. Regarding patient ID10, who is classified as M1, we added few lines to comment on this aspect (lines 298-300). Considering that we did not include a discussion on the clinical characteristic of M0 patients, we decided not to add the same comment for patient ID9. Nevertheless, the hearing thresholds of all the patients are reported in the supplementary materials. 

  1. Goiter is more common than hypothyroidism in EVA patients and is the key phenotype in distinguishing PDS from DFNB4. Fifteen subjects in this cohort age more than 10 years, but only 2 received thyroid sonography. What are the criteria of goiter/thyroid enlargement adopted in this study? The limited data of thyroid sonography made difficulties in diagnosis PDS or non-syndromic EVA.

We thank the Reviewer for highlighting this aspect. Our Institute protocol for deaf patients with EVA malformation establishes that they undergo periodic thyroid function tests (i.e., serum thyroid stimulating hormone, TSH, serum free thyroxine, T4, and serum free triiodothyronine, T3, levels). An ultrasound is required whenever altered levels are found on the blood test or if thyroid clinical examination poses a doubt on the presence of goiter or nodules. We agree with the Reviewer on the necessity to collect more thyroid sonography data and we are considering to change this protocol. Thus, we added some comments regarding these points in the manuscript (lines 113-118). As regards the criteria of goiter/thyroid enlargement adopted, we consider that in healthy adults without iodine deficiency, a normal thyroid gland is approximately 4 to 4.8 by 1 to 1.8 by 0.8 to 1.6 cm in size, with a mean volume of 8 to 10 mL (range 3 to 20 mL). Thyroid enlargement above these approximate normal measurements will be considered a goiter (Berghout et al. 1987; Maravall et al. 2004).

  1. What are the audiologic features of M0 subjects besides milder hearing loss threshold? Did they present progressive and fluctuating hearing levels?

We would like to acknowledge the Reviewer for the comment. Progression and/or fluctuation have been reported for eight out of 14 M0 patients. This lack of consistency further supports the hypothesis of different genetic contributions to the manifestation of the disease. Thus, in some cases, the HL of M0 patients might be due to mutations located in intronic regions of the SLC26A4 gene or in other genes not previously linked to PDS. We agree with the Reviewer that this is an important consideration, thus we modified the manuscript accordingly (lines 469-473).

  1. In result part, the nucleotide change of c.704A>T in SLC26A4 does not lead to p.Q235R in aminoacidic change. (Page 8, line 24)

We apologize and acknowledge the Reviewer for highlighting the mistake. We corrected the typing error and the manuscript accordingly (Table 2, lines 244 and 269-270).

  1. Page 8, table 2. Did the genetic diagnosis of ID22 subject was homozygous c.1489G>A? Did the subject belong to M2 instead of M1 group?

We thank the Reviewer and apologize for the typing error. Patient ID22 is homozygous for the c.1489G>A variant and indeed she belong to the M2 group. Thus, we corrected table 2.

  1. The pathogenicity of MYO5C should be further clarified. And more evidence of the association between MYO5C and HL is needed.

We thank the Reviewer for the comment. We definitely agree that additional studies are needed to confirm the involvement of MYO5C in the etiopathogenesis of hearing loss. However, given the high genetic heterogeneity of the disease and the difficulties in finding independent families carrying mutations in the same gene, we thought it could be worth mentioning in this paper. Sharing such findings with the scientific community can help identify additional patients with the same phenotype, providing other evidence of the pathogenic role of the new candidate gene.

In this light, we modified the text (lines 484-487) underlying the putative role of MYO5C and the need for further studies and/or the identification of other individuals with the same phenotype and carriers of MYO5C variants to prove its involvement in causing hearing loss.

We hope the revised version of our manuscript will be now suitable for acceptance by Genes.

Best regards,

Reviewer 2 Report

In this manuscript, Tesolin et al. reported their results of WES analysis about patients clinically diagnosed with Pendred Syndrome. They reported that homozygous mutations of the PDS gene could be detected in about 20% of patients, while only one or no mutations can be detected in other patients. They discussed these diversities in the manuscript. The construction of the manuscript is excellent. However, they should add a discussion to the manuscript.

Major point)

  1. As shown in the abstract (L25), one of their study's main results is that only a single mutation can be detected by deep investigation, including CEVA haplotype, in clinically diagnosed patients with Pendred syndrome. They should add two or three paragraphs and emphasize this point in the discussion session based on their clinical results and previously reported molecular biological features of Pendred syndrome. For example, previous molecular studies using human induced pluripotent stems cells from Pendred syndrome patients and genome-editing technology by TALEN suggested that even a single allele mutation of SLC26A4 might cause the disease in a specific genetic background. This discussion would be helpful for the readers.

Minor point)

  1. Several columns in Table.1 and 2 are busy. Especially, "Genetic Findings SLC26A4" in table 1 and "SLC26A4 cDNA change" in table 2 are too busy to understand at a glance. If possible, please add horizontal lines to separate each patient's results.

Author Response

Reviewer #2 (Comments to the Author):

In this manuscript, Tesolin et al. reported their results of WES analysis about patients clinically diagnosed with Pendred Syndrome. They reported that homozygous mutations of the PDS gene could be detected in about 20% of patients, while only one or no mutations can be detected in other patients. They discussed these diversities in the manuscript. The construction of the manuscript is excellent. However, they should add a discussion to the manuscript.

Major point)

As shown in the abstract (L25), one of their study's main results is that only a single mutation can be detected by deep investigation, including CEVA haplotype, in clinically diagnosed patients with Pendred syndrome. They should add two or three paragraphs and emphasize this point in the discussion session based on their clinical results and previously reported molecular biological features of Pendred syndrome. For example, previous molecular studies using human induced pluripotent stems cells from Pendred syndrome patients and genome-editing technology by TALEN suggested that even a single allele mutation of SLC26A4 might cause the disease in a specific genetic background. This discussion would be helpful for the readers.

We thank the Reviewer for the suggestion. We amended the manuscript accordingly. In particular, we added on the discussion some paragraphs to discuss more into details the possible explanations to the lack of the second allele in M1 patients (lines 416-438).

Minor point)

Several columns in Table.1 and 2 are busy. Especially, "Genetic Findings SLC26A4" in table 1 and "SLC26A4 cDNA change" in table 2 are too busy to understand at a glance. If possible, please add horizontal lines to separate each patient's results.

We thank the Reviewer for the suggestion. We modified the manuscript accordingly (Table 1 and Table 2). In particular we added horizontal lines to separate each patient's information and simplified the genetic findings by substituting the terms homozygous/heterozygous with the abbreviations Hom/Het or the classification M2, M1 and M0.

We hope the revised version of our manuscript will be now suitable for acceptance by Genes.

Best regards,

Round 2

Reviewer 2 Report

All my previous concerns have been adequately approached.